# Identification and characterization of *N9*-methyltransferase involved in converting caffeine into non-stimulatory theacrine in tea

Yue-Hong Zhang[1,2,3,6], Yi-Fang Li[1,2,3,6], Yongjin Wang[1,3,6], Li Tan[1,3], Zhi-Qin Cao[1,2,3], Chao Xie[1,2,3], Guo Xie[4], Hai-Biao Gong[1,2,3], Wan-Yang Sun[1,2,3], Shu-Hua Ouyang[1,2,3], Wen-Jun Duan[1,2,3], Xiaoyun Lu[1,3], Ke Ding [1,3], Hiroshi Kurihara[1,2,3], Dan Hu [1,2,3✉], Zhi-Min Zhang [1,3✉], Ikuro Abe [5✉] & Rong-Rong He [1,2,3✉]

Caffeine is a major component of xanthine alkaloids and commonly consumed in many popular beverages. Due to its occasional side effects, reduction of caffeine in a natural way is of great importance and economic significance. Recent studies reveal that caffeine can be converted into non-stimulatory theacrine in the rare tea plant *Camellia assamica* var. *kucha* (Kucha), which involves oxidation at the *C*8 and methylation at the *N*9 positions of caffeine. However, the underlying molecular mechanism remains unclear. Here, we identify the theacrine synthase CkTcS from Kucha, which possesses novel *N9*-methyltransferase activity using 1,3,7-trimethyluric acid but not caffeine as a substrate, confirming that *C*8 oxidation takes place prior to *N*9-methylation. The crystal structure of the CkTcS complex reveals the key residues that are required for the *N*9-methylation, providing insights into how caffeine *N*-methyltransferases in tea plants have evolved to catalyze regioselective *N*-methylation through fine tuning of their active sites. These results may guide the future development of decaffeinated drinks.

[1] International Cooperative Laboratory of Traditional Chinese Medicine Modernization and Innovative Drug Development of Chinese Ministry of Education (MOE), College of Pharmacy, Jinan University, Guangzhou, China. [2] Guangdong Engineering Research Center of Chinese Medicine & Disease Susceptibility, College of Pharmacy, Jinan University, Guangzhou, China. [3] Guangdong Province Key Laboratory of Pharmacodynamic Constituents of TCM and New Drugs Research, College of Pharmacy, Jinan University, Guangzhou, China. [4] Zhongshan Institute, University of Electronic Science and Technology of China, 528402 Zhongshan, China. [5] Graduate School of Pharmaceutical Sciences, The University of Tokyo, Tokyo 113-0033, Japan. [6] These authors contributed equally: Yue-Hong Zhang, Yi-Fang Li, Yongjin Wang. ✉email: thudan@jnu.edu.cn; 1363210756@163.com; abei@mol.f.u-tokyo.ac.jp; rongronghe@jnu.edu.cn

Caffeine is well-known as a central nervous system stimulant and a major xanthine alkaloid component in many popular beverages, especially coffee (*Coffea arabica* and *Coffea canephora*) and tea (*Camellia sinensis*). Extensive studies have well established the main biosynthetic pathway of caffeine from xanthosine (**1**), which proceeds through 7-methylxanthine (**2**) and theobromine (**3**) to yield caffeine (**4**) (Fig. 1)[1,2]. At least three different *N*-methyltransferases are required to successively add methyl groups on the *N*7, *N*3, and *N*1- positions, with *S*-adenosyl-L-methionine (SAM) as the methyl group donor[3,4].

The pharmacological actions of caffeine are believed to be mediated via non-selectively antagonizing the $A_1$ and $A_{2A}$ adenosine receptors in the central nervous system as well as peripheral tissues of the cardiovascular, respiratory, renal, and immune systems[5]. Consumption of caffeine may lead to occasional adverse effects, which include increased blood pressure, tremor, heart disease, anxiety, and gastrointestinal disturbances[6]. Awareness of these side effects has increased the demand for decaffeinated coffee and tea. However, producing a caffeine-free tea or coffee in a natural way that retains all the flavor and potential health benefit has proven to be challenging. A naturally decaffeinated *arabica* coffee species has been discovered, which has a mutatied caffeine synthase gene[7]. Alternatively, studies on the tea plant *Camellia assamica* var. *kucha* revealed that it removes caffeine in its leaves by converting caffeine into theacrine (**6**) (Fig. 1d)[8].

Theacrine is a caffeine-like xanthine alkaloid with an additional methyl group at *N*9 and a keto group at *C*8. Despite its structural similarity to caffeine, theacrine does not cause side effects such as anxiety and dehydration. Instead, it has diverse beneficial biological activities, including anti-depressive[9], sedative, and hypnotic activities[10,11], improving learning and memory[12], increasing exercise activity[13], and prevention of nonalcoholic fatty liver disease[14]. Therefore, the large-scale production of transgenic theacrine-abundant plants could be an attractive proposition, but more information is needed about the enzymes involved in the biosynthesis of theacrine.

A previous study using [14]C labeled caffeine has confirmed that theacrine is biosynthesized from caffeine, possibly initiated by oxidation at *C*8 to yield the intermediate 1,3,7-trimethyluric acid (**5**) followed by *N*9-methylation (Fig. 1d)[8]. However, the underlying molecular basis of this process remains unclear. Here, we identified three *N*-methyltransferases involved in the biosynthesis of caffeine from the leaves of Kucha. Combined with enzymatic study and kinetic analysis, we found that one of the *N*-methyltransferases, CkTcS, serves as the critical *N*9-methyltransferase, which only acts on 1,3,7-trimethyluric acid but not caffeine, indicating that *C*8-oxidation must take place prior to *N*9-methylation. The structural characterization of CkTcS identified key residues around the substrate binding pocket that are critical for *N*9-methyltransfere activity. Taken together, these studies provide key mechanistic insights into theacrine biosynthesis,

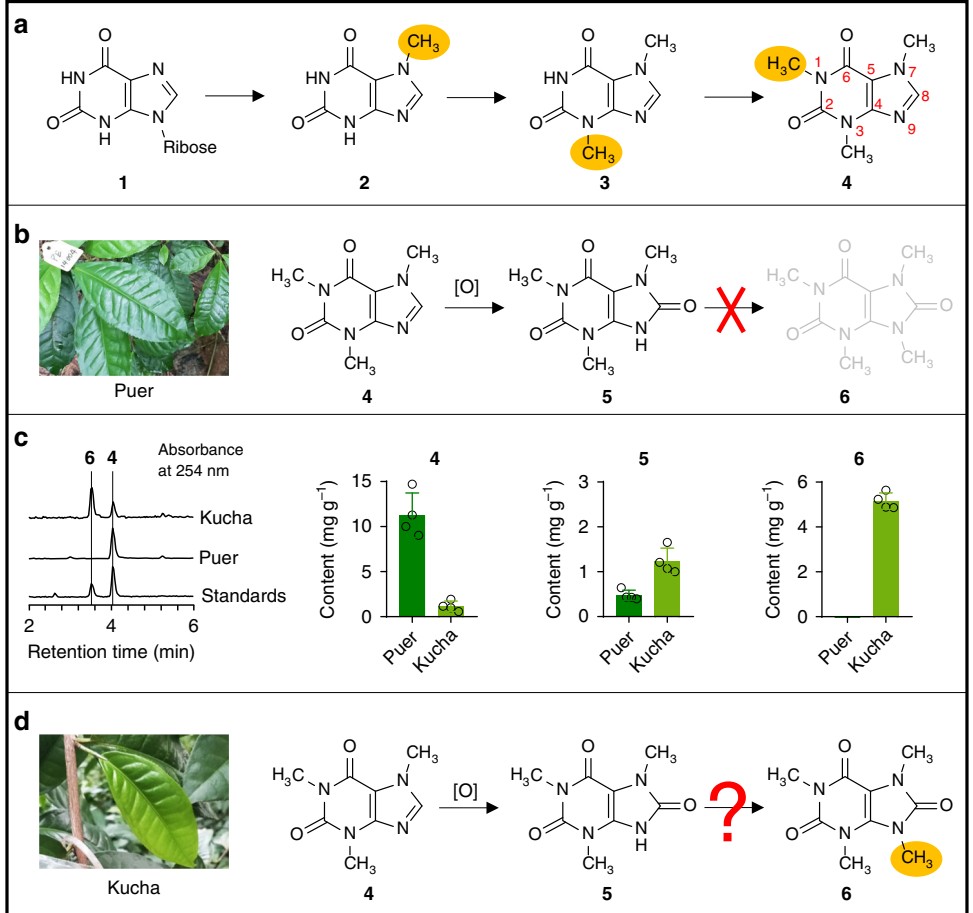

**Fig. 1 Qualitative and quantitative analysis of major xanthine alkaloids in Puer and Kucha. a** The main biosynthetic pathway of caffeine (**4**) from xanthosine (**1**). **b** Conversion of caffeine to 1,3,7-trimethyluric acid does not progress to theacrine in Puer leaves. **c** The content of caffeine (**4**), 1,3,7-trimethyluric acid (**5**) and theacrine (**6**) in Puer and Kucha leaves. The left panel shows the HPLC analysis of **4** and **6** at absorbance wavelength of 254 nm. The right three panels show the quantification of **4**, **5**, and **6** by HPLC-MS. Data represents mean ± SD (*n* = 4). The corresponding dot plots are overlaid on the figure. **d** Conversion of caffeine to 1,3,7-trimethyluric acid then to theacrine in Kucha leaves.

shedding light on a new direction for producing decaffeinated drinks.

## Results

**Cloning N-methyltransferases from Kucha.** To date, Kucha is the only plant reported to accumulate large quantities of theacrine. We used LC-MS analysis to compare the contents of caffeine (**4**), 1,3,7-trimethyluric acid (**5**), and theacrine (**6**) between Kucha and caffeine-enriched Puer tea (*Camellia sinensis* var. *assamica*) (Fig. 1c and Supplementary Fig. 1c). As expected, Kucha mainly contains theacrine and a small amount of caffeine, while Puer mainly contains caffeine without theacrine (Fig. 1c and Supplementary Fig. 1b). In addition, we detected a very small amount of caffeine metabolic intermediate **5** in both Kucha and Puer (Fig. 1c and Supplementary Fig. 1a). This result is consistent with the previous hypothesis that theacrine is synthesized from caffeine with **5** as an intermediate (Fig. 1d)[8].

The existence of the intermediate **5** in both Kucha and Puer prompted us to hypothesize that a key N9-methyltransferase may exist in Kucha, but not in Puer (Fig. 1b, d). In order to clone the N9-methyltransferase involved in theacrine biosynthesis, we extracted the total RNA from both Kucha and Puer and performed transcriptome sequencing. Blast analysis of the transcriptome data using the amino acid sequence of tea caffeine synthase 1 (TCS1) (Accession No: AB031280), a well-characterized N1, N3-methyltransferase in caffeine biosynthesis of tea, identified a partial N-terminal sequence of an N-methyltransferase (sequence ID: 35564), which is only expressed in Kucha, but not in Puer (Supplementary Fig. 2). Based on its sequence information, we designed primers to clone the potential N9-methyltransferase genes from the cDNA library derived from young Kucha leaves by PCR (Supplementary Fig. 3). As a result, three N-methyltransferases were cloned and named as CkCS, CkTbS, and CkTcS, respectively.

Phylogenetic analysis of the three N-methyltransferases identified in Kucha with those characterized in tea and coffee reveals that N-methyltransferases from tea fall within a clade distinct from those of coffee (Supplementary Fig. 4a). This is consistent with the previous proposal that N-methyltransferases from tea and coffee are evolved from two independent origins via convergent evolution[15–17]. A focused view of the phylogenetic relationships of the N-methyltransferases identified from tea showed that CkCS and CkTbS group with N1/N3 and N3-methyltransferases, respectively. In contrast, CkTcS is relatively distant to these known methyltransferases (Supplementary Fig. 4b), suggesting that it may possess a distinct function.

**CkTcS is a N9-methyltransferase.** To identify the N9-methyltransferase, we over-expressed all of them in *Escherichia coli* and purified the recombinant proteins to high purity (Fig. 2a). The enzymatic assay was performed using 1,3,7-trimethyluric acid (**5**) as a substrate, followed by HPLC analysis of the products. As shown in Fig. 2b, CkTcS exhibits a significant N9-methylation activity for converting **5** to theacrine (**6**), whereas only a miniscule amount of **6** was detected in the reactions of CkCS and CkTbS, suggesting that CkTcS is the target N9-methyltransferase. Notably, we did not detect any product when using caffeine (**4**) as a substrate (Fig. 2d, Supplementary Fig. 5a, b), confirming that oxidation at *C*8 of caffeine must take place prior to N9-methylation.

To further verify the N9-methylation activity of CkTcS, kinetic analysis of the three N-methyltransferases was performed. We initially measured the $K_m$ parameter of CkTcS for SAM to determine the saturating SAM concentration. At the saturating concentration of 50 μM substrate **5**, the $K_m$ value for SAM is about 109.50 μM (Supplementary Fig. 6a and Supplementary

Table 1). Based on this, a saturating SAM concentration of 1.5 mM was used in the subsequent kinetic experiments. Under these conditions, we measured the initial rates of the three methyltransferases at different substrate concentrations ranging from 0 to 2500 μM (Fig. 2c). Fitting the initial rate data to the Michaelis-Menten equation allowed us to derive kinetic parameters. Consistent with the enzymatic assay, CkCS and CkTbS show very weak N9-methyltransferase activity, with the $k_{cat}/K_m$ value of $1.68\,s^{-1}M^{-1}$ and $1.11\,s^{-1}M^{-1}$, respectively. In contrast, CkTcS has a much higher $k_{cat}/K_m$ value of $2440.17\,s^{-1}M^{-1}$, which is about 1500 times that of CkCS, and more than 2000 times that of CkTbS (Fig. 2c and Supplementary Table 1). These results indicate that CkTcS is a potent N9-methyltransferase.

In addition, the activity of CkTcS as well as CkCS and CkTbS towards other substrates including xanthosine (**1**), 7-methylxanthine (**2**), and theobromine (**3**) were also investigated. CkTcS shows very weak N3 and N1 methylation activities (Fig. 2d). This was further confirmed by kinetic analysis (Supplementary Fig. 6b and Supplementary Table 1) and time-course comparison (Supplementary Fig. 6c). On the other hand, CkCS and CkTbS exhibit notable N1/N3 (Supplementary Fig. 5a) and N3 (Supplementary Fig. 5b) methyltransferase activities, respectively, which is consistent with the phylogenetic analysis (Supplementary Fig. 4)

The accumulation of theacrine in Kucha but not in Puer was first thought to be caused by the deficiency of CkTcS gene in Puer. However, an N-methyltransferase gene with identical amino acid sequence to CkTcS was also cloned from Puer using the same primers, suggesting an alternative reason for theacrine accumulation in Kucha. We then compared the expression of CkTcS between Puer and Kucha. As a result, we found that the expression of CkTcS in Kucha was significantly higher than that in Puer (Fig. 3), suggesting that the significant expression of the CkTcS is the main reason for the accumulation of theacrine in Kucha.

**Crystal structure of CkTcS.** The structures of two N-methyltransferases catalyzing N7 (XMT) and N3, N1 (DXMT) methylation in *Coffea canephora* have been reported[18], but there is still no structure of N-methyltransferase derived from tea plants. To explain the substrate specificity of CkTcS, we determined the crystal structure of CkTcS in complex with 1,3,7-trimethyluric acid (**5**), in the presence of S-adenosyl-L-homocysteine (SAH), a byproduct of the cofactor SAM, at a resolution of 3.14 Å (Fig. 4, Supplementary Fig. 7 and Supplementary Table 2). Like the coffee methyltransferases, CkTcS exists as a homodimer, and this dimeric arrangement is preserved in the crystal lattice, with the dimer interface composed predominantly of hydrophobic interactions in the core region surrounded by hydrogen bonding interactions. The structure of CkTcS complex adopts a similar fold, SAH and substrate binding pockets to that of XMT and DXMT, with a backbone root-mean-square deviation (RMSD) of 1.24-1.32 Å over 251 Cα atoms (Fig. 4b). Each monomer of the CkTcS complex consists of a core SAM-dependent methyltransferase domain and a short α-helical cap domain, both of which are involved in the binding of SAH and **5** (Supplementary Fig. 7). The major structural difference between the coffee and CkTcS N-methyltransferases occurs on a helix (residues 236-255 in CkTcS), which contributes to the formation of the substrate binding pocket. In CkTcS, this helix is bent with the end close to the substrate moved away from the pocket (Fig. 4b).

**The 1,3,7-trimethyluric acid binding pocket.** The 1,3,7-trimethyluric acid is enveloped in the active site through both hydrogen bonding and van der Waals interactions, with N-9 approximate to the thioether moiety of SAH (Supplementary

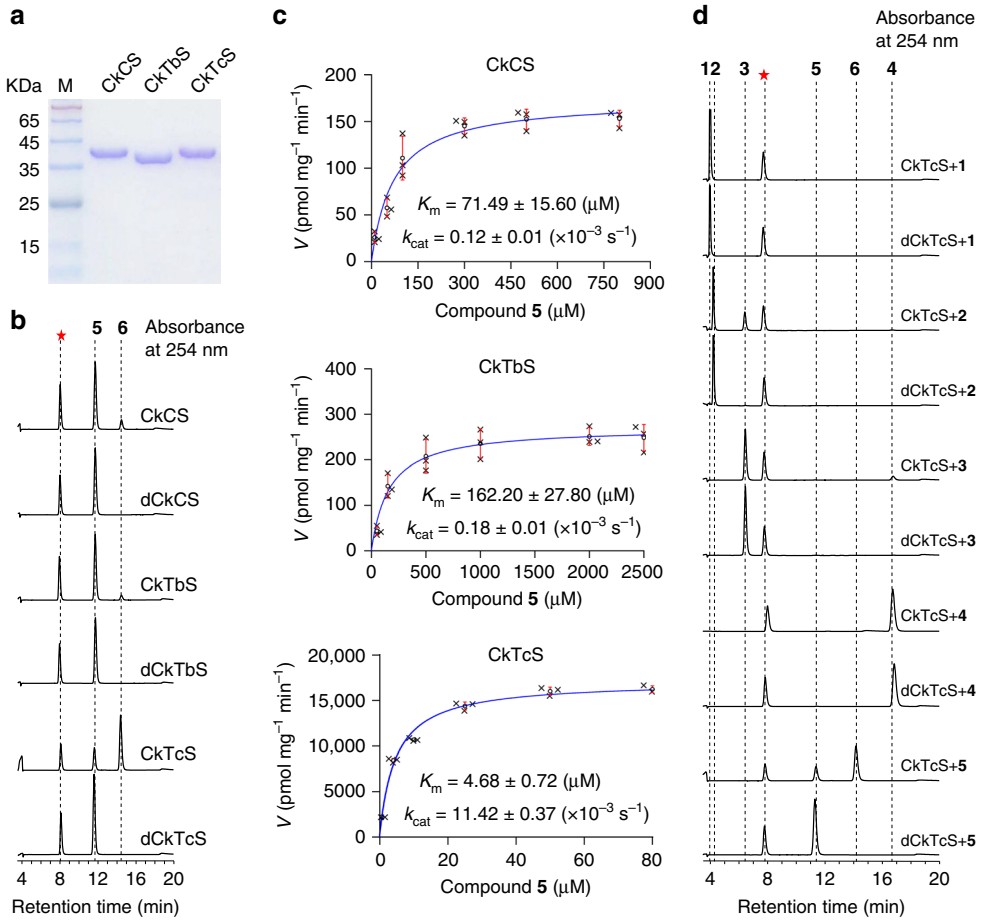

**Fig. 2 Identification of *N*9-methyltransferase involved in converting caffeine into theacrine in tea leaves. a** SDS-PAGE analysis of the recombinant CkCS, CkTbS, and CkTcS proteins. **b** In vitro *N*9-methyltransferase activity analysis of CkCS, CkTbS and CkTcS. HPLC analysis of in vitro reaction products using 1,3,7-trimethyluric acid (**5**) as a substrate with CkCS and denatured CkCS (dCkCS), CkTbS and denatured CkTbS (dCkTbS), CkTcS and denatured CkTcS (dCkTcS). The absorbance wavelength was set at 254 nm. **c** Steady state kinetic analysis of CkCS, CkTbS, and CkTcS using 1,3,7-trimethyluric acid (**5**) as a substrate. Initial velocities are shown as cycles and represented as mean ± SD (*n* = 3). The corresponding dot plots are overlaid on the figure. The blue line represents the nonlinear least-squares fit of the initial velocities versus 1,3,7-trimethyluric acid concentration to the hyperbolic Michaelis-Menten equation. Kinetic parameters were determined at a saturating concentration of 1.5 mM SAM. **d** In vitro *N*-methyltransferase activity of CkTcS and dCkTcS towards xanthosine (**1**), 7-methylxanthine (**2**), theobromine (**3**), caffeine (**4**), and 1,3,7-trimethyluric acid (**5**). The absorbance wavelength was set at 254 nm. In all HPLC chromatograms, a red asterisk indicates an impurity compound from the SAM reagent.

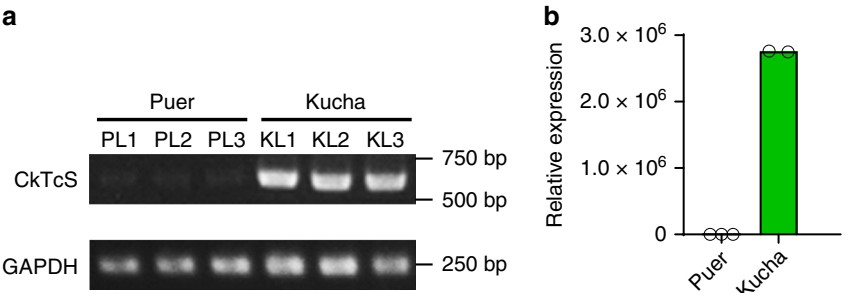

**Fig. 3 Transcriptional level analysis of CkTcS gene in Kucha and Puer. a** Qualitative analysis of CkTcS and GAPDH expression in Puer and Kucha. PL1, PL2, PL3, and KL1, KL2, KL3 represent three different leaves from Pure tea plant and Kucha tea plant, respectively. **b** Quantitative analysis of CkTcS and GAPDH expression in Puer and Kucha by quantitative real-time PCR. The gene expression in each leaf was analyzed at three replicates. The PCR was run for 40 cycles. Expression of CkTcS relative to GAPDH in three Pure leaves versus two Kucha leaves is shown.

Fig. 7). Residues surrounding the binding pocket include Met-15, Tyr-24 from the cap domain and Phe-30, Thr-31, Tyr-157, His-160, Trp-161, Arg-226, Ile-241, Trp-242, Cys-270, Ile-318, and Phe-322 from the methyltransferase domain (Fig. 4c, d). Some of these positions are commonly used by tea *N*-methyltransferases, coffee *N*-methyltransferases and even far related salicylic acid carboxyl methyltransferases (SAMT)[19], especially Met-15, Tyr-24, Try-157, Trp-161, and Ile-318, which are sequentially or

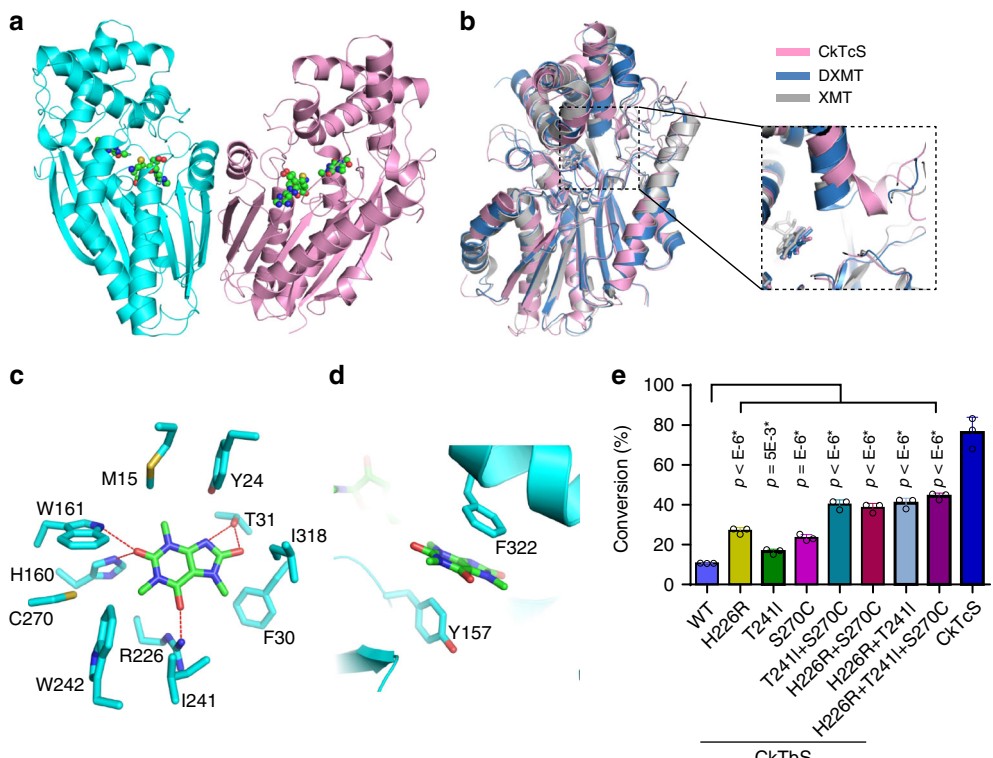

**Fig. 4 Crystal structure of CkTcS. a** Crystal structure of CkTcS dimer. SAH and 1,3,7-trimethyluric acid (**5**) are shown in ball-and-stick representation. **b** Structural overlap of CkTcS (pink) with DXMT (blue) and XMT (grey), with the major structural difference around the substrate binding pocket between CkTcS, DXMT, and XMT highlighted. **c** A close-up view of the CkTcS-1,3,7-trimethyluric acid interactions. The hydrogen bonds are shown in red dashed lines. **d** A close-up view of the π–π-stacking interactions. **e** In vitro methylation assay of wild-type (WT) or mutants of CkTbS using 1,3,7-trimethyluric acid as substrate, with CkTcS as positive control. Data represent mean ± SD (n = 3). The corresponding dot plots are overlaid on the figure. The experiment was repeated twice. Differences were assessed statistically by two-tailed Student's t-test, $^{****}P < 0.0001$, $^{*}P < 0.05$.

functionally conserved (Supplementary Fig. 8). These residues may form the primary interactions modulating the three-dimensional architecture of the substrate binding pocket.

In the CkTcS structure, **5** makes a total of five hydrogen bonds to the protein, with all the distances between the hydrogen-bond-forming atoms less than 3.0 Å. The O-6 carbonyl group of **5** is recognized by the Nε groups of Arg-226 through hydrogen-bonds; the side-chain of Trp-161 and His-160 form direct hydrogen bonds with the O-2 carbonyl group of **5** (Fig. 4c). Particularly, the hydroxyl group of Thr-31 forms hydrogen bonds with both N-9 and O-8. The O-8 position is the only distinction between caffeine and **5**, and it is critical for **5** to serve as substrate of CkTcS. The Thr-31-O-8 hydrogen bond may stabilize the O-8 carbonyl group in an iminol tautomeric state, which could facilitate attack of SAM for N9-methylation. The O-8 carbonyl group also interacts with the side-chain of Ile-318 through van der Waals interaction (Fig. 4c). Additionally, Phe-322 and Tyr-157 together sandwich the purine ring of **5** through a π–π-stacking interaction (Fig. 4d).

**N9-methylation activity of CkTcS**. We next asked why CkTcS prefers **5**, given the fact that it shares a sequence identity >90% with CkTbS and CkCS. We tried to solve the structure of CkTbS and CkCS in the presence of SAH and their corresponding substrates, but only crystallized CkTbS. Unfortunately, the N-terminal cap domain of CkTbS is disordered due to crystal packing. We finally solved the structure of CkTbS in apo form (Supplementary Fig. 9). However, when overlapping the CkTbS and CkTcS structures, most residues lining the substrate binding

pockets are in the same orientation (Supplementary Fig. 9), suggesting that substrate binding does not induce large conformational changes in the methyltransferase domain. The substrate specificity of CkTcS proteins might be determined by minor variable residues which subtly tailor the chemical features of the core catalytic scaffold. We first analyzed the residues surrounding the O-8 position. Both Phe-30 and Thr-31 are well preserved, but Ile-318 in CkTcS is replaced by a Met-318. However, the conformation of Met-318 is flexible, and it could potentially contribute a similar van der Waals interaction (Supplementary Fig. 9b). These suggest that CkTcS did not evolve a specific mechanism to distinguish **5** from other molecules on this position, even though the O-8 mediated interactions are critical for N9-methylation.

Further observation identified three sequence-variable residues around the pocket: CkTcS_Arg-226 (CkTbS_His-226, CkCS_Arg-225), CkTcS_Ile-241 (CkTbS_Thr-241, CkCS_Thr-240), and CkTcS_Cys-270 (CkTbS_Ser-270, CkCS_Ser-269) (Supplementary Figs. 8 and 9). Except for Arg-226, both Ile-241 and Cys-270 are not involved in direct interaction with **5**, and their distances to **5** are about 4.6 Å. Particularly, of the three residues, two are different between CkTcS and CkCS, three between CkTcS and CkTbS. Our kinetic study has indicated that CkCS has a higher $k_{cat}/K_m$ value compared to that of CkTbS (Fig. 2c and Supplementary Table 1). Consistently, CkTbS H226R mutation led to an apparent increase of N9-methylation activity towards **5** (Fig. 4e). Taken together, these results suggest that Arg-226 plays an active role in binding of **5** and positioning **5** properly for N9-methylation. However, Arg-226 alone is not enough: Ile-241 and Cys-270 are also indispensable. In CkTbS and CkCS, Ile-241 is

replaced by a threonine, and Cys-270 by a serine, both of which contain hydrophilic side-chains that may introduce extra direct or water-mediated hydrogen-bond and influence substrate binding. Single mutation of T241I or C270S on CkTbS led to a moderate increase in reaction products, while the double and triple mutations further increased the N-methyltransferase activity to almost half of that of CkTcS (Fig. 4e). Additionally, the equivalent residues of Ile-241 and Cys-270 in the structures of DXMT (Ser237 and Ile-266) and XMT (Ala-238 and Vle-267) are also involved in substrate discrimination[18]. Taken together, these results indicated that the N9-methylation activity of CkTcS depends on two layers of interaction: the hydrogen bond between Thr-31 and the O-8 carbonyl group to stabilize an iminol tautomeric state, and the combinatorial effort of three side-chains to accurately position 5.

### Convergent evolution of tea and coffee N-methyltransferases.

Genomic sequencing studies have revealed a convergent evolution of N-methyltransferases from tea and coffee plants[20]. We found tea N-methyltransferases position substrates with some different strategies from coffee N-methyltransferases. One striking difference is their aromatic ring recognition. In DXMT, the purine ring of theobromine is clamped through a hydrophobic-π-stacking interaction, with the perpendicular Tyr-157 forming a π-stacking interaction on one face, whereas Ile-332 makes a hydrophobic interaction on the opposite face (Supplementary Fig. 10). Similar interactions are also observed in the structure of XMT and even SAMT[18,19]. Sequence alignment indicates that the Ile-332 position is occupied by a conserved phenylalanine residue in tea plant N-methyltransferases, whereas it is either an isoleucine or a valine residue in other N-methyltransferases (Supplementary Fig. 8). This suggests that the π-π-stacking interaction might be a unique way of substrate recognition by tea plant N-methyltransferases. Another obvious difference is that there are two sequence-conserved tyrosine residues in N-methyltransferases of coffee plants that are involved in constructing the pocket. In the structures of DXMT (Tyr-333 and Tyr-368) and XMT (Tyr-321 and Tyr-356), both of the two tyrosine residues are involved in direct interaction with the substrates through hydrophobic interaction or hydrogen-bond[18], but their counterparts in tea N-methyltransferases are usually serine or threonine residues (Supplementary Fig. 8). In CkTcS, those residues are too distant from 5 (Supplementary Fig. 10). These results clearly indicated that the caffeine N-methyltransferases from tea and coffee adopt a distinct substrate binding strategy even though they possess the same methylation activity, which provides further evidence for the convergent evolution of the N-methyltransferases from tea and coffee in terms of their enzyme structure.

### Discussion

N-methyltransferases play key roles in the pathway of caffeine biosynthesis. Extensive studies have well established that caffeine biosynthesis relies on methylation on the N7, N1, and N3 positions by corresponding N-methyltransferases. This study presents, to our best knowledge, the first N9-methyltransferase in caffeine metabolism, which will expand our understanding of this important protein family.

Genomic sequencing studies have revealed a convergent evolution of the N-methyltransferases among tea and coffee plants. Usually, the sequence variety between N-methyltransferases with different catalytic activities from the same origin is less apparent than that of N-methyltransferases with the same catalytic activities from different origins. This is evident even though N-methyltransferases from both origins share a similar structural fold, substrate binding pocket and dimeric conformation. To recognize different substrates and achieve high methylation-

position specificity, these enzymes subtly manipulate the orientation of the substrate through the combined effort of a set of residues lining the binding pocket.

Our results confirmed that the conversion of caffeine to theacrine happens with oxidation at the C8 position first, followed by methylation at the N9 position. So far, the enzyme responsible for C8 oxidation is still unclear. The detection of 5 in both Kucha and Puer suggests that this enzyme might be ubiquitous in the tea plant, and the N9-methylation is the key step to consume 5 and produce theacrine. Surprisingly, we also cloned the CkTcS gene from Puer, which produces no theacrine at all, but its expression level is strictly controlled.

Traditionally, the level of caffeine in the plants is decreased through the low activity of caffeine biosynthetic genes or the rapid degradation of caffeine[21]. The identification of N9-methyltransferase could guide mutagenesis work on existing functionally redundant N-methyltransferases in some tea plants to convert caffeine to theacrine, which has diverse beneficial biological activities but not the side effects of caffeine. In fact, theacrine[22] and 1,3,7-trimethyluric acid were also isolated from coffee (Supplementary Fig. 11), suggesting that a similar conversion of caffeine to theacrine may exist in some coffee plants, too[23]. The greatest benefit of this strategy is that the whole caffeine biosynthesis pathway is still intact in the plants, which may help to keep the quality and aroma of the coffee beans. Our study therefore points a new direction for production of caffeine-deficient drinks.

## Methods

**Chemicals.** S-adenosyl-L-methionine (SAM) was purchased from Sangon Biotech. S-adenosyl-L-homocysteine (SAH) was purchased from Sigma-Aldrich. Xanthosine, theobromine and caffeine were obtained from Wako Pure Chemical Industries, Ltd. 7-methylxanthine was purchased from Target Molecule. 1,3,7-trimethyluric acid and theacrine were supplied by Shanghai Better-In Pharmaceutical Technology Co., Ltd.

**Plant materials.** Young leaves from the tea plants Puer (*Camellia sinensis* var. *assamica*) and Kucha (*Camellia assamica* var. *kucha*) were collected at the experimental farm of the University of Electronic Science and Technology, Zhongshan City, Guangdong Province, China. Samples were collected in the Spring. The leaves were frozen in dry ice immediately after harvest and stored at −80 °C.

**Analysis of major Xanthine alkaloids of tea and coffee.** Tea leaves were initially dried in an oven at 85 °C for about 1.5 h. The dried tea leaves and fresh coffee beans were ground into fine powder using a ceramic mortar. The resulting powders of tea leaves (50 mg) and fresh coffee beans (1.0 g) were incubated in distilled water at 95 °C for 40 min, respectively. The precipitate was removed by centrifugation at $12,240 \times g$ for 10 min after cooling. The supernatant was filtered by a 0.22 μM nylon filter before injection. Analysis was performed on a liquid chromatography tandem with mass spectrometry (LC/MS) system using a Dionex Ultimate 3000 HPLC system coupled on-line to a Q-Exactive Hybrid Quadrupole-Orbitrap Mass Spectrometer (Thermo Fisher Scientific, San Jose, CA) using an HSS T3 column (2.1 × 100 mm, 1.8 μm, Waters Acquity). The analysis was performed using gradient solvents (A) 0.1% formic acid in water (0.1%, v/v) and (B) acetonitrile at a flow rate of 0.4 mL min⁻¹. The gradient was as follows: 8% B at 0 min, 8% B at 1.0 min, 13% B at 1.5 min, 15% B at 4.5 min, 17% B at 6.0 min, 27% B at 8.0 min, 60% B at 13.0 min, 80% B at 15.0 min, 80% B at 17.0 min. The column was maintained at 40 °C. The wavelength was set at 254 nm. 2 μL of sample solvent was subjected to LC/MS. Analysis was performed in both negative and positive ion mode at a resolution of 70,000 for full MS scan and 17,500 for MS² scan in data-dependent mode. The scan range for MS analysis was m/z 100–1500 with a maximum injection time of 100 ms using 1 microscan. An isolation window of 1.0 Da was set for MS² scans. Capillary spray voltage was set at 3.5 kV and −2.5 kV in positive and negative ion modes, respectively. Capillary temperature was set at 320 °C. The S-lens RF level was set to 60. The standard solutions of caffeine (12.2–194.0 μg mL⁻¹), 1,3,7-trimethyluric acid (6.1-195.3 ng mL⁻¹), and theacrine (28.0–896.0 μg mL⁻¹) were prepared to quantitate the content of xanthine alkaloids in tea plants. In addition, the standard solutions of 1,3,7–trimethyluric acid (6.6–105.0 ng mL⁻¹) and theacrine (14.0–224.0 ng mL⁻¹) were prepared to determinate the content of xanthion alkaloids in fresh coffee beans.

**RNA extraction and cDNA preparation**. The tea leaves were broken into pieces and ground in liquid nitrogen with a mortar. Total RNA was extracted following the manufacturer's protocol of the RNeasy plant mini kit (QIAGEN). The first strand cDNA was prepared with a PrimeScript™ 1st Strand cDNA Synthesis Kit (TaKaRa).

**Transcriptome sequencing**. Transcriptome sequencing in this study was performed at the Novogene Bioinformatics Institute (Novogene, Beijing, China). Briefly, mRNA was purified from the total RNA of tea leaves and used to construct sequencing libraries using a NEBNext® Ultra™ RNA Library Prep Kit for Illumina® (NEB, USA) according to the manufacturer's recommendations. The index codes were added to attribute sequences to each sample. After the qualities of sequencing libraries were confirmed on the Agilent Bioanalyzer 2100 system, clustering of the index-coded samples was performed on a cBot Cluster Generation System using TruSeq PE Cluster Kit v3-cBot-HS (Illumia, San Diego, CA, USA) according to the manufacturer's instructions. The library preparations were then sequenced on an Illumina Hiseq 2500 platform (Illumina, San Diego, CA, USA) and paired-end reads were generated. Transcriptome assembly was accomplished based on the left. fq and right.fq using Trinity with min_kmer_cov set to 2 by default and all other parameters set default. Gene function was annotated based on the following databases: NCBI non-redundant Protein (NR), NCBI non-redundant nucleotide (NT), Protein family (Pfam), Clusters of Orthologous Groups of proteins (KOG/COG), Swiss-prot, KEGG Ortholog (KO) and Gene Ontology (GO). Gene expression levels were estimated by RNA-Seq by Expectation Maximization (RSEM) for each sample.

**Cloning of the *N*-methyltransferase gene sequence**. Transcriptome sequencing was completed by Beijing Nuohezhiyuan Technology Service Co, Ltd. A pair of oligonucleotide primers, 5′-ATGGAGCTAGCTACTAGGG-3′ and 5′-CTATC-CAACAATCTTGGAAAGC-3′, were designed based on transcriptome data. The forward primer contains the start codon region of 35564 sequence and the reverse primer contains the stop codon region of 35562 sequence, respectively. Polymerase chain reaction (PCR) was performed using Prime STAR® HS DNA Polymerase (TaKaRa) in 50 μL of reaction mixture containing cDNA and the primers mentioned above. The PCR protocol includes denaturation at 98 °C for 2 min, 35 cycles of denaturation at 98 °C for 10 s, annealing at different temperatures for 15 s, and extension at 72 °C for 70 s, followed by a final extension at 72 °C for 5 min. The amplified DNA fragments were purified by polyacrylamide gel electrophoresis and were subcloned into the pBluescript II SK(+) vector. Subsequently, the resulting plasmids were transformed into *Escherichia coli* (DH5α). Twenty clones were randomly selected in each transformation. DNA sequencing was performed by Shanghai Shenggong Bioengineering Co., Ltd. Nucleotide sequences were analyzed with the CLC sequence Viewer 7 program.

**Protein purification**. The cDNAs encoding CkCS, CkTbS and CkTcS were cloned into a pRSF-Duet vector with an *N*-terminal His₆-SUMO tag. The plasmids were verified by sequencing and then transformed into BL21(DE3) cells. When the cell density reached an OD₆₀₀ of 0.6, protein expression was induced using 0.4 mM isopropyl β-D-1-thiogalactopyranoside (IPTG) at 16 °C overnight. Cells were harvested, resuspended and lysed in buffer A [50 mM Tris-HCl (pH 8.0), 25 mM imidazole, 1 M NaCl, and 1 mM PMSF]. The recombinant proteins were purified using a Nickel column and eluted by buffer B [25 mM Tris-HCl (pH 8.0), 250 mM imidazole, 100 mM NaCl]. The eluted proteins were incubated with ubiquitin-like-protease 1 (ULP1) on ice to cleave the His₆-SUMO tag, and the tag-free proteins were further fractioned by ion-exchange (Q, GE Healthcare) and size-exclusion (Superdex 200, GE Healthcare) chromatography. Purified proteins were stored in a buffer containing 25 mM Tris (pH 7.5), 100 mM NaCl, 5 mM DTT at a concentration of 30 mg mL⁻¹.

**In vitro assay for *N*-Methyltransferase**. The *N*-methylation in vitro assays were performed in triplicate at 27 °C for 16 h[24]. Briefly, a 100 μL reaction mixture contained 8 μg purified recombinant protein (CkCS, CkTbS, and CkTcS), 1.5 mM SAM, 400 μM substrate (**1**, **2**, **3**, **4**, and **5**), 100 mM Tris-HCl (pH 7.0), and 200 μM MgCl₂. The reactions were quenched by equal volumes of methanol. Boiled proteins were used as negative controls. The reaction mixture was centrifuged to remove precipitate, and then the supernatant was analyzed using the same LC system on a C18 reverse-phase column (4.6 × 250 mm, 5 μm, COSMOSIL). The analysis was performed using gradient solvents (A) 0.1% formic acid in water (0.1%, v/v) and (B) acetonitrile at a flow rate of 1.0 mL min⁻¹. The gradient was set as follows: 8% B at 0 min, 8% B at 5.0 min, 12% B at 20.0 min, 90% B at 20.1 min, 90% B at 23.0 min. The column temperature was maintained at 40 °C. 20 μL of the supernatant was subjected to the HPLC-UV system. The wavelength was set at 254 nm. Reaction products were identified by comparing retention times and ultraviolet spectra with those of authentic standards. The peaks of the substrates and products were integrated to calculate the conversion efficiency.

To compare the methylation activity of CkTcS between theobromine (**3**) and 1,3,7- trimethyluric acid (**5**), reactions were performed in a volume of 200 μL containing 10 μM purified protein CkTcS, 1.5 mM SAM, 500 μM substrate (**3** and **5**), 100 mM Tris-HCl (pH 7.0) and 200 μM MgCl₂. The mixtures were incubated at

27 °C for varying times (15, 60, 120, 240, 480 min). After centrifugation, 20 μL supernatant was subjected to HPLC-UV system and isocratically eluted at 8% B for 18 min using the same solvent system mentioned above.

**Kinetic analysis**. Compounds **3** and **6** were quantified by HPLC-MS and SAM was quantified by HPLC-UV. All were calculated using the external standard method with their corresponding calibration curves as shown in Supplementary Fig. 12.

The kinetic parameters of CkTcS with SAM were measured by consumption of SAM. The reactions were performed in a final volume of 200 μL containing 1 μM purified enzyme CkTcS, 100 mM Tris-HCl (pH 7.0), 200 μM MgCl₂, 50 μM compound **5** and varying concentrations (10–600 μM) of SAM. The mixtures were incubated at 27 °C for 3 min and then terminated by adding 200 μL methanol. After centrifugation, 20 μL of supernatant was subjected to HPLC-UV and was separated on a reverse-phase column (4.6 × 250 mm, 5 μm, COSMOSIL) with the column temperature maintained at 40 °C. The analysis was performed using a binary elution program with solvents (A) 0.1% formic acid in water (0.1%, v/v) and (B) acetonitrile. The gradient was set as follows: 5% A at 0 min, 15% A at 3.5 min, 15% A at 7.0 min, 5% A at 7.5 min, 5% A at 13.0 min. The flow rate was set at 1.0 mL min⁻¹. The wavelength was set at 254 nm. The same method was used to determine the kinetic parameters of other enzymes and substrates, with minor adjustments as follows:

The kinetic parameters of CkCS with **5** were measured by formation of product **6**. 2 μM CkCS was incubated with varying concentrations of substrate **5** (10–800 μM) and 1.5 mM SAM for 10 min. Product **6** was analyzed using the HPLC-MS system.

The kinetic parameters of CkTbS with **5** were measured by formation of product **6**. 4 μM CkTbS was incubated with varying concentrations of substrate **5** (50–2500 μM) and 1.5 mM SAM for 15 min. Product **6** was analyzed using HPLC-MS system.

The kinetic parameters of CkTcS with **5** were measured by formation of product **6**. 0.2 μM CkTcS was incubated with varying concentrations substrate **5** (0.5–80 μM) and 1.5 mM SAM for 3 min. Product **6** was analyzed using the HPLC-MS system.

The kinetic parameters of CkTcS with **2** were measured by formation of product **3**. 1 μM CkTcS was incubated with varying concentrations substrate **2** (50–2500 μM) and 1.5 mM SAM for 15 min. Product **3** was analyzed by the same HPLC-MS system as mentioned above with an isocratic elution at 8% B for 13 min.

The *Km* and *Vmax* values were calculated using the Michaelis-Menten kinetics equation by nonlinear regression analysis with GraphPad Prism 6 (GraphPad Software, La Jolla, CA). Kinetics were performed in triplicate and each data point represents the mean of three independent assays with error bars representing the standard deviation (± SD).

**Transcript level analysis of CkTcS gene in Puer and Kucha**. RNAs from Kucha and Puer leaves were extracted using a RNeasy plant mini kit (Qiagen), and 2 μg of RNA was reverse transcribed to cDNA using *TransScript®* One-Step gDNA Removal and cDNA Synthesis SuperMix. Polymerase chain reaction (PCR) was performed using Prime STAR® HS DNA Polymerase (TaKaRa) in 50 μL of reaction mixture containing cDNA and the gene-specific primers, 5′-TTCAAGCTATTAA CGCAGCA-3′ and 5′-GGCTATAGCTAATAGTTCCCAAA- 3′, under the conditions of 98 °C for 2 min, 28 cycles of denaturation at 98 °C for 10 s, annealing at 55 °C for 15 s, and extension at 72 °C for 70 s, followed by a final extension at 72 °C for 5 min. The PCR reaction conditions of the internal reference gene GAPDH (primers: 5′-ATCGTTGAGGGTCTCATGAC-3′ and 5′-CACTGAGACATCGAC AGTGG-3′) were 98 °C for 2 min, 35 cycles of denaturation at 98 °C for 10 s, annealing at 50 °C for 15 s, and extension at 72 °C for 70 s, followed by a final extension at 72 °C for 5 min. The amplified CkTcS and GAPDH gene were detected by polyacrylamide gel electrophoresis. In addition, the cDNA samples were amplified on a CFXConnect™ Real-Time system (Bio-Rad, Hercules, CA, USA) in the presence of SYBR qPCR Master Mix (TOYOBO, OSAKA, JP) and specific primers described above. The mRNA expression of the CkTcS gene was calculated by the delta cycle threshold method, and the results were normalized to the reference gene GAPDH.

**Crystallization and structure determination**. For crystallization, eight residues on the *N*-terminal of CkTcS were truncated based on predicted structural disorder. CkTcS was mixed with 2 mM SAH and 5 mM 1,3,7-trimethyluric acid on ice for 30 min to form a complex. The crystals were generated by the hanging-drop diffusion method at 4 °C, from drops composed of 2 μL of complex solution and 2 μL of precipitant solution [0.1 M HEPES (pH 7.5), 1 M Sodium Citrate]. The full-length CkTbS was crystallized similarly in 0.1 M MES (pH 6.0), 0.2 M Li₂SO₄, 20% PEG4000. Glycerol was added to the precipitant solution to a final concentration of 15–20% (v/v) to cryo-protect the crystals during flash freezing in liquid nitrogen. X-ray diffraction data were collected on the beam-line BL17U1 at Shanghai Synchrotron Radiation Facility. The data were indexed, integrated and scaled by the HKL2000 package[25]. The structures were solved by molecular replacement using PHASER[26] and the DXMT structure (PDBcode: 2EFJ) as the searching model. Iterative cycles of model rebuilding and refinement were carried out using COOT (Version 0.8.1) [27] and PHENIX (Version 1.9-1692)[28], respectively. 1,3,7-tri-methyluric acid and SAH were not filled into the electron densities until the proteins were well refined. The shape of electron density in the substrate binding pocket is consistent with the structure of 1,3,7-trimethyluric acid. Potential interactions of 1,3,7-trimethyluric acid with surrounding residues were also taken into

consideration to determine its orientation. Data collection and refinement statistics are summarized in Supplementary Table 2.

**Reporting summary**. Further information on research design is available in the Nature Research Reporting Summary linked to this article.

## Data availability

The structures of CkTcS-SAH-1,3,7-trimethyluric acid and CkTbS have been deposited in the Protein Data Bank under codes 6LYH and 6LYI, respectively. The GenBank accession numbers for nucleotide sequences of CkCS, CkTbS, and CkTcS are MN163829, MN163830, and MN163831, respectively. Assembled transcriptome data are available via DRYAD at https://datadryad.org/stash/share/ZymTdk5n80wG4-H7JebWmqJFkkkIBKn PqCPzJ1xE0y0. Additional data underlying Figs. 1c, 2c, 3b, 4e and Supplementary Figures 6 and 11 are available as a separate Source Data file.

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

## Acknowledgements

We thank Dr Wei-Xi Li and Ms Xing-Qian Sun for providing the coffee materials and photos. This work was supported, in part, by National Key Research and Development Program of China (grant number 2017YFC1700404), Natural Science Foundation of China (grant numbers 81622050, 81573675, U1801284, 81673709, 81873209, 31800638, and 31870032), the Local Innovative and Research Teams Project of Guangdong Pearl River Talents Program (grant number 2017BT01Y036) and GDUPS(2019), the Guangdong Natural Science Funds for Distinguished Young Scholar (2019B151502014), Science and Technology Program of Guangzhou (grant number 201903010062), and Natural Science Foundation of Guangdong Province (grant numbers 2018A030313003 and 2017A030306004). Assistance with Scientific English was provided by Dr. L.J. Sparvero (University of Pittsburgh).

## Author contributions

R.R.H. and H.K. conceived and oversaw the project. I.A., D.H., and Z.M.Z. designed and managed the experiments. Y.H.Z. performed most of the experiments. Z.M.Z., Y.W., and L.T. constructed wild and mutant plasmids and purified proteins, performed crystallization trials, collected diffraction data, determined, and refined the structures. Z.Q.C. assisted with gene cloning and in vitro enzyme reaction experiment. C.X. assisted with vitro enzyme activity test analysis. C.X., G.X., and S.H.O. assisted with tea sample collection. W.Y.S., H.B.G., and S.H.O. assisted with qualitative and quantitative analysis of alkaloids in tea. W.J.D., assisted with the quantitative analysis of CkTcS gene expression in tea. X.L and K.D were involved in discussion of the results. Y.H.Z. and Y.F.L prepared the initial manuscript. D.H., Z.M.Z., I.A., and R.R.H. wrote the manuscript.

## Competing interests

The authors declare no competing interests.
