## [Peer Review File · Nature Communications]

Reviewers' comments:

Reviewer #1 (Remarks to the Author):

The manuscript entitled "Identification and characterization of N9-methyltransferase involved in converting caffeine into non-stimulatory theacrine in tea" by Zhang et al. report the discovery of a new N9-methyltransferase, CkTcS, in Kucha tea plants that converts 1,3,7-trimethyluric acid to theacrine. In addition they identify an N3 (CkTbS) and N1/N3 (CkCS) methyltransferase, similar to those previously identified in tea and coffee plants. Furthermore, the overexpression of CkTcS in Kucha is likely responsible for the high level of theacrine observed when compared to other species that preferentially synthesise caffeine. The authors go further by using X-ray crystallography to determine the crystal structure of CkTcS in complex with SAH and theacrine, and an apo-CkTbS one, the first xanthine alkaloid N-methyltransferases from tea to be structurally characterized. These structures confirm the hypothesis of a convergent evolution for this family of enzymes in coffee and tea plants. The structures were also used to identify potential residues important for substrate binding, which were confirmed by mutagenesis and biochemical assays.

The experiments are well described and performed, and all the results together provide a thorough characterisation of CkTcS from Kucha that could be useful in the generation of new tea and coffee plant varieties with reduced levels of caffeine. The results are new and bring an important contribution to the plant secondary metabolite field in general and xanthine alkaloid one in particular. In my opinion the key finding that CkTcS is the last enzyme in the theacrine biosynthesis pathway, and that its overexpression is the most likely reason for theacrine accumulation in Kucha is worthy of further consideration for publication in Nature Communication. However, a number of issues would have to be addressed in the current manuscript, including many grammatical errors, before further recommendation.

Major criticisms

1. Pg. 4, ln 58. The authors claim that the reason for reduced caffeine in the Ethiopian arabica strain is degradation (ref 7). This is incorrect as the paper clearly states it's most likely due to a mutated caffeine synthase gene. Please correct this very serious error.
2. Pg 5, ln 65. The authors claim theacrine is implicated in 'anti-depress', (which should be 'as an anti-depressive'), via reference 12. But reference 12 is clearly unrelated to such studies as it describes the study of a completely different plant biosynthesis pathway?
3. The fact that CkTcS did evolve to specifically recognize 1,3,7-trimethyluric acid is interesting, if expected, and consistent with the very low rate of theacrine synthesis observed with CkTbS and CkCS. However, from the text on Pg. 12 and Fig.4 legend (which reports mutants of CkTcS?) it's unclear which N-methyltransferase was selected for mutation (CkTbS presumably as is mentioned on Pg12?) to recover ~50% of the theacrine synthase conversion rate. Can the authors please correct/amend the text and Fig4 (e) legend so it's clear and consistent which protein was mutated?
4. The authors should perhaps mention that Ile-241/Thr-241 in CkTcS and CkTbS, respectively, and one of the residues important for theacrine synthase, is equivalent to Ser-237 in CcDXMT. Here, Ser-237 mediates a H-bond interaction to O6 of 7-methylxanthine as part of a substrate recognition loop in the related DXMT, XMT and SAMT enzymes. In addition, Cys-270/Ser-270 in CkTcS and CkTbS, respectively, and another of the residues important for theacrine synthase, is equivalent to Ile-266 in CcDXMT, again postulated as a potential substrate discrimination residue for 7-methylxanthine vs theobromine in coffee but not tea.
5. Out of curiosity I would also be interested to know if Kucha tea plants are more (or less) robust/productive than Puer tea plant? Is this known? This could be important from an agricultural selection process.

6. Pg.18 and 19 – Assay. Can the authors please note in the methods or HPLC figures/legends (Fig. 2 (d) and Fig. S5) the wavelength used for these chromatograms (eg. on the Y-axis: Absorbance @ X nm?). This is important for people wanting to reproduce the experimental conditions.

Minor criticisms

Pg. 3, In 30/31 'However, the molecular mechanism underlying is still unclear.' should be 'However, the underlying molecular mechanism remains unclear.' Or similar

Pg. 3, In 31 – 'CkTcS' is in bold, why?

Pg. 4, In 56 – '...health benefit is proved...' should be '...health benefit has proven...'

Pg. 4, In 57 – '...species have been...' is plural, it should be singular, ie. '...species has been...'

Pg. 4, In 58 – Reference 8 is labelled as 'recent' but this publication dates from 2002 so it's not really 'recent', 'previous' is probably a better adjective.

Pg. 4, In 59 – '...by conversing caffeine...' should be '...by converting caffeine...'

Pg. 5, In 81 – '...shedding lights on a new...' should be '...shedding light on a new...'

Pg. 6, In 84 – '...only plant to be reported that accumulates large quantity of...' should be '...only plant reported to accumulate large quantities of...'

Pg. 9, In 156 – '...by hydrogen interactions.' Should be '...by hydrogen bonding interactions.'

Pg. 9, In 257 – '...CkTcS complex adopts a similar folding, SAH and...' – should be '...CkTcS adopts a similar fold, SAH and...' Through text referred to as

Pg. 9., In 159 – '...over 251 atoms...' – Is this 251 C α atoms? Please specify the atom type.

Pg. 10, In 184/185 – '...in an iminol state , which then attacks the SAM to form methylation on N-9 position.' should be '...in an iminol tautomeric state, which could facilitate attack of SAM for N-9 methylation.'

Pg. 12, In 216 – 'However, Arg -226 along is not....' should be 'However, Arg-226 alone is not....'

Pg. 9, In 163 – '...occur on a helix which....' The helix number or preferably the residue numbers should be included here.

Pgs. 15 to 22 – Many typos and grammatical mistakes to be correct but I ran out of time to point these out. Eg:

Pg. 15, In 384

Pg. 16, In 307-308

Pg. 17, In 316 et 325

Pg. 19, In 360

Pg. 20, In 394

Pg. 21, In 414

Note - all the temperatures are shown as boxes in the methods section (at least in my version).

Supplemental figures

1. Figure S6 – In (a) the legend reports the methyltransferase domain and N-terminal cap are colored cyan and red, respectively. However, this is not the case here.
2. Figure S7 - On Pg. 15, In 274-275 the authors mention cloning the CkTcS gene from Puer, which produces no caffeine because it's expression levels are low. If the gene is identical (Pg. 8, In 141) this should also be added to Figure legend S7 to make this clear to readers.

PDB files

The CkTbS PDB deposition metrics look reasonable but the CkTcS look a little worse than I would expect, especially the sidechain outlier number. While this is probably due to the lower resolution perhaps the authors should perhaps relook at these residues. In addition the density shown for 1,3,7-trimethyluric acid in Fig. S6 is not totally unambiguous at 1.5 sigma. As this is an integral part of the manuscript's results maybe the authors should add a few more details in the methods on how this orientation was selected, positioning of N9 towards SAM?

Reviewer #2 (Remarks to the Author):

The manuscript by Zhang et al. described the identification and characterization of a novel N9 methyltransferase which is responsible for the degradation of caffeine by converting 1,3,7, trimethyluric acid to theacrine. The respective cDNA was discovered by a comparative transcript profiling approach between a low caffeine, high theacrine containing tea (*Camellia assamica* var. kucha, Kucha) and a high caffeine, low theacrine containing tea variety (*Camellia assamica* var. assamica, Puer). The enzyme exhibits strong N9-methylation specificity compared to N1, N3, and N7 methylation when using purines as substrates. It was also shown, that the other two N-methyltransferase, which they isolate, are not able to perform N9 methylation, but rather catalyze N3,1 and N3 methylation (Caffeine synthase, and theobromine synthase). Crystallization studies also revealed difference in the substrate binding pocket, which discriminate N9 methylation specificity from N7 and N3,1 methylation specificity. Remarkably, the authors managed to convert a theobromine synthase (N3 methylation) to a theacrine synthase (N9 methylation) by changing 3 amino acids in the substrate binding pocket. The results are discussed with respect to a potential genetic engineering of coffee plants with the novel theacrine synthase in order to generate caffeine-free coffee plants for the production of decaffeinated coffee. The work has been well performed, the results are clear and clearly presented, and the conclusion are well justified by the results. However, there are some issues, which should be addressed:

1. The results are discussed with the future prospect to engineer caffeine-free coffee plants by enhancing caffeine degradation. This might well be possible using the novel N9 methylating theacrine synthase. However, several things should be borne in mind:
 - a) what would be the advantage compared to already existing caffeine free coffee varieties?
 - b) do coffee plants actually possess the enzyme for 8 oxidation of caffeine to produce 1,3,7-trimethyluric acid as theacrine substrate? In Fig 1c, the authors show that both tea varieties contain 1,3,7-trimethyluric acid, which justifies the assumption that theacrine synthase is the decisive step for theacrine production in tea. However, is this also the case in coffee? Could the authors measure the 1,3,7-trimethyluric acid levels in coffee, especially in coffee beans?
2. In Fig.3 the authors show that theacrine synthase is expressed at a much higher level in the theacrine containing Kucha variety compared to the high caffeine Puer variety. However, could the authors also follow transcript levels during the development of tea plants and determine the transcript levels in different organs, and correlate these transcript levels with theacrine levels? This would strengthen the arguments that the in vivo role of theacrine N9 methyltransferase is caffeine degradation.
3. Kinetic analysis: Fig. 2 and Fig. S5: v at the y axis in Fig. 2 c should be expressed as specific

activity in nmol/ mg/min. Actually the SI unit for enzyme velocities is katal (s⁻¹). The authors should give the right units in the graph. Additionally, could the authors mention the r² values in order to estimate how well the curves fit to the data? Are there reasons why the authors only showed the curves up to a concentration of 150 μM, 1000 μM, or 10 μM? Especially since they state in the material and method part, that they used substrate concentration up to about 1000 μM. In order to accurately determine K_m values, one should also measure the velocity at saturating substrate concentrations, i.e. roughly at 10 x K_m. For substrate solubility reasons, this might be difficult for TbS, but it should be possible for the other two enzymes. The number of measurements should also be indicated. Furthermore, the authors describe that the kinetic measurements were performed at saturating SAM concentrations of 1.5 mM. Although this is probably saturating (at least compared to other methyltransferase with K_m values of about 200 μM), one should also determine SAM affinities. Also, the differences in the three NMTs to accept 1,3,7 trimethyluric acid as substrate are clearly shown in Fig.2c. However, what about the affinity of theacrine synthase for 7-Methylxanthine and Theobromine? Especially since in Fig 2d iii, there seem to be a considerable conversion of 7-Methylxanthine to Theobromine. Although it is unlikely that theacrine is not the product of theacrine synthase, could the authors provide an MS/MS spectrum of the reaction product, just for the sake of completeness?

4. How many replicates were used for the RNA seq experiments?

5. How did the authors quantify caffeine, 1,3,7 trimethyluric, and theacrine? The units of the y axis in Fig.1 c are mg/g and μg/g, respectively. However, I did not read anything about a quantification standard in the material and method section.

6. The authors found three NMT sequences (35564,35563, 35562) and finally isolated three NMTs, Theobromine synthase, Caffeine synthase, and Theacrine synthase. Could the authors indicate, which characterized NMT belongs to which NMT obtained from the RNA seq data? In Fig.S2 the meaning of the shading should also be indicated.

7. In Fig. S4, bootstrap values and a distance bar should be indicated.

We thank both reviewers for their critical review and positive comments on our work, which greatly help improve the manuscript. We tried our best to address their concerns systematically and revised the manuscript accordingly. Please find our point-by-point response to each of the reviewers' comments below.

Reviewers' comments:

Reviewer #1 (Remarks to the Author):

The manuscript entitled “Identification and characterization of N9-methyltransferase involved in converting caffeine into non-stimulatory theacrine in tea” by Zhang et al. report the discovery of a new N9-methyltransferase, CkTcS, in Kucha tea plants that converts 1,3,7-trimethyluric acid to theacrine. In addition they identify an N3 (CkTbS) and N1/N3 (CkCS) methyltransferase, similar to those previously identified in tea and coffee plants. Furthermore, the overexpression of CkTcS in Kucha is likely responsible for the high level of theacrine observed when compared to other species that preferentially synthesise caffeine. The authors go further by using X-ray crystallography to determine the crystal structure of CkTcS in complex with SAH and theacrine, and an apo-CkTbS one, the first xanthine alkaloid N-methyltransferases from tea to be structurally characterized. These structures confirm the hypothesis of a convergent evolution for this family of enzymes in coffee and tea plants. The structures were also used to identify potential residues important for substrate binding, which were confirmed by mutagenesis and biochemical assays.

The experiments are well described and performed, and all the results together provide a thorough characterization of CkTcs from Kucha that could be useful in the generation of new tea and coffee plant varieties with reduced levels of caffeine. The results are new and bring an important contribution to the plant secondary metabolite field in general and xanthine alkaloid one in particular. In my opinion the key finding that CkTcS is the last enzyme in the theacrine biosynthesis pathway, and that it's overexpression is the most likely reason for theacrine accumulation in Kucha is worthy of further consideration for publication in Nature Communication.

Response: We are grateful to the reviewer for commenting on our manuscript as “The results are new and bring an important contribution to the plant secondary metabolite field in general and xanthine alkaloid one in particular”. Our responses to the reviewer’s advice on the major and minor criticisms are listed below.

Major criticisms

1. Pg. 4, ln 58. The authors claim that the reason for reduced caffeine in the Ethiopian arabica strain is degradation (ref 7). This is incorrect as the paper clearly states it’s most likely due to a mutated caffeine synthase gene. Please correct this very serious error.

Response: We are very sorry about this error. It has been corrected in the revised manuscript. We also checked the all the other references carefully to make sure they are cited in the right way.

2. Pg 5, ln 65. The authors claim theacrine is implicated in ‘anti-depress’, (which should be ‘as an anti-depressive’), via reference 12. But reference 12 is clearly unrelated to such studies as it describes the study of a completely different plant biosynthesis pathway?

Response: We thank the reviewer for pointing this out. A wrong reference was inserted when preparing the manuscript. We have replaced it with the right one.

3. The fact that CkTcS did evolved to specifically recognize 1,3,7-trimethyluric acid is interesting, if expected, and consistent with the very low rate of theacrine synthesis observed with CkTbS and CkCS. However, from the text on Pg. 12 and Fig.4 legend (which reports mutants of CkTcS?) it’s unclear which N-methyltransferase was selected for mutation (CkTbS presumably as is mentioned on Pg12?) to recover ~50% of the theacrine synthase conversion rate. Can the authors please correct/amend the text and Fig4 (e) legend so it’s clear and consistent which protein was mutated?

Response: We have revised both the text and Fig. 4 legend to make it clear that CkTbS was selected for mutations.

4. The authors should perhaps mention that Ile-241/Thr-241 in CkTcS and CkTbs, respectively, and one of the residues important for theacrine synthase, is equivalent to Ser-237 in CcDXMT. Here, Ser-237 mediates a H-bond interaction to O6 of 7-methylxanthine as part of a substrate recognition loop in the related DXMT, XMT and SAMT enzymes. In addition, Cys-270/Ser-270 in CkTcS and CkTbs, respectively, and another of the residues important for theacrine synthase, is equivalent to Ile-266 in CcDXMT, again postulated as a potential substrate discrimination residue for 7-methylxanthine vs theobromine in coffee but not tea.

Response: Following the reviewer's suggestion, we have added "Additionally, the equivalent residues of Ile-241 and Cys-270 in the structures of DXMT (Ser237 and Ile-266) and XMT (Ala-238 and Ile-267) are also involved in substrate discrimination" in the main text (Page 11).

5. Out of curiosity I would also be interested to know if Kucha tea plants are more (or less) robust/productive than Puer tea plant? Is this known? This could be important from an agricultural selection process.

Response: We have not compared the productivity between Kucha and Puer. However, based on the size of Kucha plants, we believe kucha is as productive as Puer.

6. Pg.18 and 19 – Assay. Can the authors please note in the methods or HPLC figures/legends (Fig. 2 (d) and Fig. S5) the wavelength used for these chromatograms (eg. on the Y-axis: Absorbance @ X nm?). This is important for people wanting to reproduce the experimental conditions.

Response: The wavelength used for chromatograms is 254 nm. We have added it on the figures, and noted it in the methods.

Minor criticisms

Pg. 3, ln 30/31 ‘However, the molecular mechanism underlying is still unclear.’ should be ‘However, the underlying molecular mechanism remains unclear.’ Or similar

Response: We have revised the manuscript as suggested.

Pg. 3, ln 31 – ‘CkTcS’ is in bold, why?

Response: We have changed it to base font.

Pg. 4, ln 56 – ‘...health benefit is proved...’ should be ‘...health benefit has proven...’

Response: We have revised the manuscript as suggested.

Pg. 4, ln 57 – ‘...species have been...’ is plural, it should be singular, ie. ‘...species has been...’

Response: We have revised the manuscript as suggested.

Pg. 4, ln 58 – Reference 8 is labelled as ‘recent’ but this publication dates from 2002 so it’s not really ‘recent’, ‘previous’ is probably a better adjective.

Response: We have revised the manuscript as suggested.

Pg. 4, ln 59 – ‘...by conversing caffeine...’ should be ‘...by converting caffeine...’

Response: We have revised the manuscript as suggested.

Pg. 5, ln 81 – ‘...shedding lights on a new...’ should be ‘...shedding light on a new...’

Response: We have revised the manuscript as suggested.

Pg. 6, ln 84 – ‘...only plant to be reported that accumulates large quantity of...’ should be ‘...only plant reported to accumulate large quantities of...’

Response: We have revised the manuscript as suggested.

Pg. 9, ln 156 – ‘...by hydrogen interactions.’ Should be ‘...by hydrogen bonding interactions.’

Response: We have revised the manuscript as suggested.

Pg. 9, ln 257 – ‘...CkTcS complex adopts a similar folding, SAH and...’ – should be ‘...CkTcS adopts a similar fold, SAH and...’ Through text referred to as

Response: We have revised the manuscript as suggested.

Pg. 9., ln 159 – ‘...over 251 atoms...’ – Is this 251 Ca atoms? Please specify the atom type.

Response: The backbone root-mean-square deviation (RMSD) was calculated based on Ca atoms of the overlapped structures. We have revised the manuscript as suggested.

Pg. 10, ln 184/185 – ‘...in an iminol state , which then attacks the SAM to form methylation on N-9 position.’ should be ‘...in an iminol tautomeric state, which could facilitate attack of SAM for N-9 methylation.’

Response: We have revised the manuscript as suggested.

Pg. 12, ln 216 – ‘However, Arg -226 along is not....’ should be ‘However, Arg-226 alone is not....’

Response: We have corrected this error in the manuscript.

Pg. 9, ln 163 – ‘...occur on a helix which....’ The helix number or preferably the residue numbers should be included here.

Response: The residue numbers (236-255 in CkTcS) have been included in the revised the manuscript.

Pgs. 15 to 22 – Many typos and grammatical mistakes to be correct but I ran out of time to point these out. Eg:

Pg. 15, ln 384

Pg. 16, ln 307-308

Pg. 17, ln 316 et 325

Pg. 19, ln 360

Pg. 20, ln 394

Pg. 21, ln 414

Response: We have carefully checked the rest part of the manuscript, especially the method part, and corrected all the errors pointed out by the reviewer, as well the ones we found out. Moreover, the manuscript has been scientifically edited by Dr. L.J. Sparvero from the University of Pittsburgh.

Supplemental figures

1. Figure S6 – In (a) the legend reports the methyltransferase domain and *N*-terminal cap are colored cyan and red, respectively. However, this is not the case here.

Response: The Fig. S6 was revised as Fig. S7. In the Fig. S7, the methyltransferase domain and the *N*-terminal cap are colored as reported in the legend. However, the *N*-terminal cap is a short loop, making it difficult to be distinguished. We have labeled both the methyltransferase domain and the *N*-terminal cap in a revised Fig. S7.

2. Figure S7 - On Pg. 15, ln 274-275 the authors mention cloning the CkTcS gene from Puer, which produces no caffeine because it's expression levels are low. If the gene is identical (Pg. 8, ln 141) this should also be added to Figure legend S7 to make this clear to readers.

Response: The Fig. S7 was revised as Fig. S8. As suggested, we have added this to the Figure legend of Fig. S8.

PDB files

The CkTbS PDB deposition metrics look reasonable but the CkTcS look a little worse than I would expect, especially the sidechain outlier number. While this is

probably due to the lower resolution perhaps the authors should perhaps relook at these residues. In addition the density shown for 1,3,7-trimethyluric acid in Fig. S6 is not totally unambiguous at 1.5 sigma. As this is an integral part of the manuscript's results maybe the authors should add a few more details in the methods on how this orientation was selected, positioning of N9 towards SAM?

Response: We understand the reviewer's concern. The resolution of the CkTcS complex is indeed not as good as that of the CkTbS. We tried very hard to improve the quality of the data by screening hundreds of new crystals or by merging data from several crystals, but failed. Therefore, we had to focus on this 3.15 Å data. We removed some side-chains with poor electron density, improved the Ramachandran plot, and performed rounds of refinement. Now the side-chain outlier number is much less, as shown in the updated validation report.

1,3,7-trimethyluric acid and SAH were not filled into the electron densities until the proteins were well defined. Actually, we didn't have much trouble to position 1,3,7-trimethyluric acid. The shape of electron density in the substrate binding pocket is consistent to the structure of 1,3,7-trimethyluric acid. Potential interactions of 1,3,7-trimethyluric acid with surrounding residues, especially R226 and T31, are also taken into consideration to determine its orientation. As suggested, we have added this information in the methods.

Reviewer #2 (Remarks to the Author):

The manuscript by Zhang et al. described the identification and characterization of a novel N9 methyltransferase which is responsible for the degradation of caffeine by converting 1,3,7, trimethyluric acid to theacrine. The respective cDNA was discovered by a comparative transcript profiling approach between a low caffeine, high theacrine containing tea (*Camellia assamica* var. kucha, Kucha) and a high

caffeine, low theacrine containing tea variety (*Camellia assamica* var. *assamica*, Puer). The enzyme exhibits strong N9-methylation specificity compared to N1, N3, and N7 methylation when using purines as substrates. It was also shown, that the other two N-methyltransferase, which they isolate, are not able to perform N9 methylation, but rather catalyze N3,1 and N3 methylation (Caffeine synthase, and theobromine synthase). Crystallization studies also revealed difference in the substrate binding pocket, which discriminate N9 methylation specificity from N7 and N3,1 methylation specificity. Remarkably, the authors managed to convert a theobromine synthase (N3 methylation) to a theacrine synthase (N9 methylation) by changing 3 amino acids in the substrate binding pocket. The results are discussed with respect to a potential genetic engineering of coffee plants with the novel theacrine synthase in order to generate caffeine-free coffee plants for the production of decaffeinated coffee. The work has been well performed, the results are clear and clearly presented, and the conclusion are well justified by the results.

Response: We thank the review for his/her positive comments on our manuscript. Our responses to the reviewer's questions are listed below.

1. The results are discussed with the future prospect to engineer caffeine-free coffee plants by enhancing caffeine degradation. This might well be possible using the novel N9 methylating theacrine synthase. However, several things should be borne in mind:
 - a) what would be the advantage compared to already existing caffeine free coffee varieties?

Response: In the existing caffeine-deficient coffee plants, the level of caffeine is usually decreased through the low activity of caffeine biosynthetic genes or the rapid degradation of caffeine. However, most of these plants are not suitable for commercial exploitation because of the poor quality and bitter taste of the resulting beverage and the low productivity of the trees

(*Euphytica*, 2008, 164, 133-142). The *N*⁹-methyltransferase identified in the present study may shed light on a new direction to produce decaffeinated plants. By genetic engineering work on existing functionally redundant *N*-methyltransferases, caffeine can be converted into the non-stimulatory theacrine, while at the same time the whole caffeine biosynthesis pathway is still intact in the plants, which may help to keep the quality and aroma of the coffee beans.

b) do coffee plants actually possess the enzyme for 8 oxidation of caffeine to produce 1,3,7-trimethyluric acid as theacrine substrate? In Fig 1c, the authors show that both tea varieties contain 1,3,7-trimethyluric acid, which justifies the assumption that theacrine synthase is the decisive step for theacrine production in tea. However, is this also the case in coffee? Could the authors measure the 1,3,7-trimethyluric acid levels in coffee, especially in coffee beans?

Figure S11. Measurement of 1,3,7-trimethyluric acid and theacrine in coffee plant. **a**, the picture of the coffee beans of *coffee arabica*. **b**, The content of 1,3,7-trimethyluric acid and theacrine in the beans of *coffee arabica*.

Response: We appreciate the reviewer for raising this issue. 1,3,7-trimethyluric acid was isolated from the leaves of some coffee species, but whether it exists in the beans was not reported (*Plant Physiol.* 1983, 73, 961-964). We could not get coffee beans of those species. Instead, we obtained some leaves and beans of *Coffea arabica L*, which are grown in Yunan Province of China. By using high performance MS, we indeed detected both 1,3,7-trimethyluric acid and theacrine in the fresh beans, but not in the leaves (Fig. S11). The content of

1,3,7-trimethyluric acid in the beans is at a similar level to that in Kucha and Puer, but the theacrine is less than 1/3000 to that in Kucha. This result indicated that a similar conversion from caffeine to theacrine may also exist in some coffee plants.

However, we agree with the reviewer that it should be cautious when discussing the use of *N9*-methyltransferase in coffee plants. There is still no solid evidence to support the idea that caffeine is converted into theacrine in the same way as that in tea plant. We rewrote this part as “The identification of *N9*-methyltransferase could guide mutagenesis work on existing functionally redundant *N*-methyltransferases in some tea plants to convert caffeine to theacrine, which has diverse beneficial biological activities but not the side effects of caffeine. In fact, 1,3,7-trimethyluric acid and theacrine was also isolated from some coffee species, suggesting that a similar conversion may exist in some coffee plants, too. Our study therefore points a new direction for production of caffeine-deficient drinks.” (Page 14).

2. In Fig.3 the authors show that theacrine synthase is expressed at a much higher level in the theacrine containing Kucha variety compared to the high caffeine Puer variety. However, could the authors also follow transcript levels during the development of tea plants and determine the transcript levels in different organs, and correlate these transcript levels with theacrine levels? This would strengthen the arguments that the *in vivo* role of theacrine *N9* methyltransferase is caffeine degradation.

Response: The reviewer raised a valid point. To correlate the transcript levels of *N9*-methyltransferase with the theacrine levels in time and organ scale will greatly strengthen our conclusion. However, we perceive that these experiments could not be achieved in a short time frame. It is winter in China now, and the work is going take a whole year. We will perform these experiments in the future studies.

3. Kinetic analysis: Fig. 2 and Fig. S5: v at the y axis in Fig. 2 c should be expressed as specific activity in nmol/ mg/min. Actually the SI unit for enzyme velocities is katal (s⁻¹). The authors should give the right units in the graph.

Response: According to your suggestions, we have changed the unit of v at the y axis to pmol/mg/min in the kinetic analysis (Fig. 2c, Fig. S6a and Fig. S6b), and checked the units in all figures throughout the article.

Additionally, could the authors mention the R^2 values in order to estimate how well the curves fit to the data?

Response: We have added the R^2 values of all the curves in Table S1.

Are there reasons why the authors only showed the curves up to a concentration of 150 μ M, 1000 μ M, or 10 μ M? Especially since they state in the material and method part, that they used substrate concentration up to about 1000 μ M. In order to accurately determine K_m values, one should also measure the velocity at saturating substrate concentrations, i.e. roughly at 10 x K_m . For substrate solubility reasons, this might be difficult for TbS, but it should be possible for the other two enzymes. The number of measurements should also be indicated.

Response: According to your requests, we have reperformed these experiments at saturating substrate concentrations (approximately 10 x K_m). The detail of the methods has been described in methods section of kinetic analysis (pages 19-20) and the results for kinetic parameters are shown in Fig. 2c, Fig. S6 and Table S1. Kinetics were performed in triplicate and each data point represents the mean of the three independent assays with error bars representing the standard deviation (\pm SD) (page 20).

Furthermore, the authors describe that the kinetic measurements were performed at saturating SAM concentrations of 1.5 mM. Although this is probably saturating (at

least compared to other methyltransferase with K_m values of about 200 μM), one should also determine SAM affinities.

Response: We have measured the affinity of CkTcS for SAM, and determine its K_m value (109.50 μM) (Fig. S6a). This result thus confirmed that our kinetic experiments were performed in a saturating SAM condition (1.5 mM SAM). (page 6-7).

Also, the differences in the three NMTs to accept 1,3,7-trimethyluric acid as substrate are clearly shown in Fig.2c. However, what about the affinity of theacrine synthase for 7-Methylxanthine and Theobromine? Especially since in Fig 2d iii, there seem to be a considerable conversion of 7-Methylxanthine to Theobromine.

Response: According to your suggestion, we have measured the kinetic parameters of CkTcS to 7-Methylxanthine (Fig. S6b and Table S1). But for Theobromine, because of its low solubility and a poor substrate of CkTcS, we could not accurately measure its kinetic parameters. Instead, we compared the methylation activity of CkTcS between theobromine and 1,3,7 trimethyluric acid via a time course analysis (Fig. S6c). All the above results indicated that CkTcS a specific *N9*-methyltransferase..

Although it is unlikely that theacrine is not the product of theacrine synthase, could the authors provide an MS/MS spectrum of the reaction product, just for the sake of completeness?

Response: According to your suggestion, we have provided the MS/MS spectrum of the reaction product in Fig. S1, and compared with the authentic compound. We thus confirmed that the product is theacrine.

4. How many replicates were used for the RNA seq experiments?

Response: The purpose of the RNA seq experiment is not for quantitative analysis, but only for us to obtain the primer sequence for cloning the specific *N9*-methyltransferase. Thus, we only performed the RNA-seq experiments once.

5. How did the authors quantify caffeine, 1,3,7-trimethyluric acid, and theacrine? The units of the y axis in Fig.1 c are mg/g and $\mu\text{g/g}$, respectively. However, I did not read anything about a quantification standard in the material and method section.

Response: According to your request, we have described the quantitative procedure in the method section in more detail (page 15-16, page 19). The quantification standard curves were also added in Fig. S1 and Fig. S12.

6. The authors found three NMT sequences (35564,35563, 35562) and finally isolated three NMTs, Theobromine synthase, Caffeine synthase, and Theacrine synthase. Could the authors indicate, which characterized NMT belongs to which NMT obtained from the RNA seq data? In Fig.S2 the meaning of the shading should also be indicated.

Response: Among the three NMT sequences, only 35564 was found to derive from CkTcS gene (Fig. S2), while 35563 and 35562 are not belong to our characterized NMTs in this study. This suggests that 35562 and 35563 may be from other NMTs in Kucha. The shading regions in Fig. S2 indicates the conserved amino acids among these NMTs.

7. In Fig. S4, bootstrap values and a distance bar should be indicated.

Response: We have added bootstrap values and a distance bar in Fig. S4.

REVIEWERS' COMMENTS:

Reviewer #1 (Remarks to the Author):

The authors have addressed most of my concerns in the revised manuscript and so I'm happy to recommend publication after the authors submit the correct PDB validation report (see below). Additionally, the new experimental results further validate the original conclusions and together with the grammatical corrections improve the overall quality of the manuscript. I also noted a few minor corrections that should be addressed as well.

My main concern is that while the preliminary validation report for the CkTcS ligand bound structure is better the 'official' PDB validation reports are missing and should be submitted to the journal. The current ones are 'preliminary' reports, which clearly state they should not be submitted to journals. Please resubmit the correct ones as the ligand validation for CkTcS is missing in the preliminary one, and this is important.

Secondly, please report the Rpim value (or similar) instead of Rmerge, which is not a valid metric to show anymore. A redundancy-independent (Rrim) or precision merging (Rpim) value is better. Apologies, I should have asked for this in the original report.

Thirdly, the wavelength used for HPLC runs is mentioned only once in the methods section (in the kinetic analysis), but not the previous 'Qualitative and quantitative analysis etc.' (Pg. 15) or 'In vitro etc' assay sections. Please add the wavelength used here and in following Figure legends (Figure 2b and d, Supplementary Figure 1a and b, and Supplementary Figure 5a and b). It's not obvious what the '254 nm' in the figures serves, absorbance @ 254nm most likely.

Minor corrections.

Pg. 5, In 1: 'We used by LC-MS analysis...' should be 'We used LC-MS analysis'

Pg. 9, In 15-16: '...which are sequential or functional conserved (Supplementary Fig. 8).' Should be '...which are sequentially and functionally conserved (Supplementary Fig. 8).'

Pg. 9, In 20: Sentence should start with a capital ('the'...should be 'The')

Pg. 10, In 16: '...large conformation change...' should be '...large conformational changes...'

Pg. 10, In 18: '...tailor the chemical feature...' should be '...tailor the chemical features..' (I think).

Pg. 10, In 21: '...and it potentially could contribute..' is probably better as '...and it could potentially contribute...'

Pg. 11, In 10: Here the authors state that CkCS has 'a much higher Kcat/Km compared to CkTbs'. I would argue that 0.01 vs 0.07 mM⁻¹ min⁻¹ is not really significant enough to warrant the use of 'much', and would recommend it be deleted in this context

Pg. 11, In 18: '...to influence the way of substrate binding' better and simpler as '...and influence substrate binding.'

Pg. 12, In 5: '...and the combination effort of a triad to...' should be '...and the combinatorial effort of three side chains to...'

Reviewer #2 (Remarks to the Author):

Review Zhang et al., revision

The manuscript considerably improved compared to the first version. There are only minor corrections which should be done to further improve it. In order to emphasize the importance of the MS, I suggest to include some aspects of the response to reviewer #2 about the benefits of using the conversion of caffeine to theacrine in order to generate decaffeinated coffee in the introduction or in the discussion, if space constraints are not a limiting factor

1) There several figures, in which error bars are shown, but which are not defined as either standard deviations or standard errors. Similarly, if showing statistics, the number of replicates

should be indicated. Although it is mentioned in the Material and Method section, this information should also be included in the respective legends

2) The writing could still be improved. For example in line 57: ...has been discovered to decaffeinate by mutating..., this does not sound like a proper sentence, or line 87: We used by HPLC.... There are several mistakes which should be corrected.

3) Supplemental Figure 4: Could the authors define the species abbreviations used in the phylogenetic tree by indicating the full species name in the legend? Also, they should indicate that the values are bootstrap values with the addition of how often the tree was calculated. Additionally, could they provide a similarity matrix of the amino acid identity between the tea NMTs?

4) I think, the authors should stick to the SI units for the enzyme kinetics, i.e. k_{cat} is s^{-1} and k_{cat}/K_m should be $mM^{-1} s^{-1}$ or $M^{-1} s^{-1}$

5) For all kinetic data, whether shown as a graph or as a table: The authors should always indicate also in the legend the concentration of the constant co-substrate with which they recorded the kinetic data of a certain substrate

6) Figure 3: I could not see the error bars, although it is stated in the legend. May be it is too small to see? Although stated in the material and method section, the authors should indicate that the relative expression levels in the figure refer to the GAPDH expression. Also the primer sequences for the determination of GAPDH expression should be shown.

7) Is there a confusion in Figure 4d)? According to the text and Figure S8 F157 should be F322 and Y322 should be Y157. For figure 4e) the substrate should be named in the legend, which was probably compound 5

8) Figure S8: According to the legend, the Histidine at position 160 in the YSVHW motif should also be highlighted by a green dot, since it is conserved in all MTs. The residues which are exchanged in Figure 4e) should also be highlighted by a dot. This would make it easier for the reader to follow the mutation analysis.

9) Figure S9: Please indicate which compound is present in b). It is probably compound 5, but the oxygen at position 8 is hardly visible

10) Line 225: I guess the comparison is for compound 5. However, I do not see that CkCS has a much higher k_{cat}/K_m value compared to CkTbS. It is higher, but 0.1 vs 0.07 is not a huge difference. One should make a more moderate statement.

Reviewers' Comments

We thank both reviewers again for their critical comments on our work. We have revised the manuscript accordingly to address their concerns. Please find our point-by-point response to each of the reviewers' comments below.

Reviewer #1 (Remarks to the Author):

My main concern is that the while the preliminary validation report for the CkTcS ligand bound structure is better the 'official' PDB validation reports are missing and should be submitted to the journal. The current ones are 'preliminary' reports, which clearly state they should not be submitted to journals. Please resubmit the correct ones as the ligand validation for CkTcS is missing in the preliminary one, and this is important.

Response: We understand the review's concern. The structures have been deposited in the PDB bank, and the final validation reports containing the ligand validation have been provided.

Secondly, please report the R_{pim} value (or similar) instead of R_{merge} , which is not a valid metric to show anymore. A redundancy-independent (R_{rim}) or precision merging (R_{pim}) value is better. Apologies, I should have asked for this in the original report.

Response: R_{pim} value has been provided as suggested in the Supplementary Table 2.

Thirdly, the wavelength used for HPLC runs is mentioned only once in the methods section (in the kinetic analysis), but not the previous 'Qualitative and quantitative analysis etc.' (Pg. 15) or 'In vitro etc' assay sections. Please add the wavelength used here and in following Figure legends (Figure 2b and d, Supplementary Figure 1a and b, and Supplementary Figure 5a and b). It's not obvious what the '254 nm' in the figures serves, absorbance @ 254nm most likely.

Response: We have added the wavelength in the methods section, and also the figure legends of Figure 2b and d, and Supplementary Figure 5a and b. But Supplementary Figure 1a and b are not obtained from the HPLC-UV analysis, but the HPLC-MS analysis. "absorbance at 254 nm" has been added in the corresponding figures.

Minor corrections.

Pg. 5, In 1: 'We used by LC-MS analysis...' should be 'We used LC-MS analysis'

Response: We have revised the manuscript as suggested.

Pg. 9, In 15-16: '...which are sequential or functional conserved (Supplementary Fig. 8).' Should be '...which are sequentially and functionally conserved (Supplementary Fig. 8).'

Response: We have revised the manuscript as suggested.

Pg. 9, In 20: Sentence should start with a capital ('the'...should be 'The')

Response: We have revised the manuscript as suggested.

Pg. 10, In 16: '...large conformation change...' should be '...large conformational changes...'

Response: We have revised the manuscript as suggested.

Pg. 10, In 18: ‘...tailor the chemical feature...’ should be ‘...tailor the chemical features..’ (I think)

Response: We have revised the manuscript as suggested.

Pg. 10, In 21: ‘...and it potentially could contribute..’ is probably better as ‘...and it could potentially contribute...’

Response: We have revised the manuscript as suggested.

Pg. 11, In 10: Here the authors state that CkCS has ‘a much higher K_{cat}/K_m compared to CkTbs’. I would argue that 0.01 vs 0.07 $\text{mM}^{-1} \text{min}^{-1}$ is not really significant enough to warrant the use of ‘much’, and would recommend it be deleted in this context.

Response: We have removed the word “much” from the sentence.

Pg. 11, In 18: ‘...to influence the way of substrate binding’ better and simpler as ‘...and influence substrate binding.’

Response: We have revised the manuscript as suggested.

Pg. 12, In 5: ‘...and the combination effort of a triad to...’ should be ‘...and the combinatorial effort of three side chains to....’

Response: We have revised the manuscript as suggested.

Reviewer #2 (Remarks to the Author):

The manuscript considerably improved compared to the first version. There are only minor corrections which should be done to further improve it. In order to emphasize the importance of the MS, I suggest to include some aspects of the response to reviewer #2 about the benefits of using the conversion of caffeine to theacrine in order to generate decaffeinated coffee in the

introduction or in the discussion, if space constraints are not a limiting factor

Response: Following the reviewer's suggestion, we have added "The greatest benefit of this strategy is that the whole caffeine biosynthesis pathway is still intact in the plants, which may help to keep the quality and aroma of the coffee beans" in the discussion.

1) There several figures, in which error bars are shown, but which are not defined as either standard deviations or standard errors. Similarly, if showing statistics, the number of replicates should be indicated. Although it is mentioned in the Material and Method section, this information should also be included in the respective legends.

Response: The error bars in the figures (Figure 1c, 2c, 4e, and supplementary Figure 6, 11) represent the standard deviation. The number of replicates is also indicated. We have added this information in the figure legends of all related figures.

2) The writing could still be improved. For example in line 57:has been discovered to decaffeinate by mutating....., this does not sound like a proper sentence, or line 87: We used by HPLC..... There are several mistakes which should be corrected.

Response: we have revised these as suggested.

3) Supplemental Figure 4: Could the authors define the species abbreviations used in the phylogenetic tree by indicating the full species name in the legend? Also, they should indicate that the values are bootstrap values with the addition of how often the tree was calculated. Additionally, could they provide a similarity matrix of the amino acid identity between the tea NMTs?

Response: Following the review's suggestion, we have provided the full species names in the legend. And the numbers on the nodes of phylogenetic tree are indicated as bootstrap values, which represent the

phylogenetic confidence of the tree topology. We also provided a similarity matrix of the amino acid identity between the tea NMTs in Supplementary Figure 4b.

4) I think, the authors should stick to the SI units for the enzyme kinetics, i.e. k_{cat} is s^{-1} and k_{cat}/K_m should be $mM^{-1} s^{-1}$ or $M^{-1} s^{-1}$

Response: We have used SI units for the enzyme kinetics in Figure 2c, Supplementary Figure 6 and Supplementary Table 1.

5) For all kinetic data, whether shown as a graph or as a table: The authors should always indicate also in the legend the concentration of the constant co-substrate with which they recorded the kinetic data of a certain substrate

Response: We have indicated the concentration of the constant co-substrate in the legends of Figure 2c and Supplementary Figure 6.

6) Figure 3: I could not see the error bars, although it is stated in the legend. May be it is too small to see? Although stated in the material and method section, the authors should indicated that the relative expression levels in the figure refer to the GAPDH expression. Also the primer sequences for the determination of GAPDH expression should be shown.

Response: In Fig. 3b, the gene expression of CkTcS from one of the Kucha leaves has been excluded from analysis, because it is far from those of the other two leaves, thus we did not calculate S.D. or perform t-tests when $n = 2$. In the figure legend, we have indicated that the expression of CkTcS is relative to GAPDH, and the primer sequences for the determination of GAPDH expression are provided in the method section.

7) Is there a confusion in Figure 4d)? According to the text and Figure S8 F157 should be F322 and Y322 should be Y157. For figure 4e) the substrate should

be named in the legend, which was probably compound 5

Response: We have revised the error in Figure 4d and named the substrate in the legend, which is indeed compound 5.

8) Figure S8: According to the legend, the Histidine at position 160 in the YSVHW motif should also be highlighted by a green dot, since it is conserved in all MTs. The residues which are exchanged in Figure 4e) should also be highlighted by a dot. This would it make easier for the reader to follow the mutation analysis.

Response: We have revised Figure S8 as suggested.

9) Figure S9: Please indicate which compound is present in b). It is probably compound 5, but the oxygen at position 8 is hardly visible

Response: The compound is compound 5. We have labeled it in the new figures.

10) Line 225: I guess the comparison is for compound 5. However, I do not see that CkCS has a much higher k_{cat}/K_m value compared to CkTbS. It is higher, but 0.1 vs 0.07 is not a huge difference. One should make a more moderate statement.

Response: We have removed “much” from the sentence.